# Phase Behavior of NR/PMMA Semi-IPNs and Development of Porous Structures

**DOI:** 10.3390/polym15061353

**Published:** 2023-03-08

**Authors:** Jacob John, Damir Klepac, Mia Kurek, Mario Ščetar, Kata Galić, Srećko Valić, Sabu Thomas, Anitha Pius

**Affiliations:** 1Department of Chemistry, Gandhigram Rural Institute, Dindigul 624302, Tamil Nadu, India; 2Department of Medical Chemistry, Biochemistry and Clinical Chemistry, Faculty of Medicine, University of Rijeka, Braće Branchetta 20, HR-51000 Rijeka, Croatia; 3Centre for Micro- and Nanosciences and Technologies, University of Rijeka, Radmile Matejčić 2, HR-51000 Rijeka, Croatia; 4Faculty of Food Technology and Biotechnology, University of Zagreb, Pierottijeva 6, HR-10000 Zagreb, Croatia; 5Rudjer Bošković Institute, Bijenička 54, HR-10000 Zagreb, Croatia; 6School of Chemical Sciences, Mahatma Gandhi University, Kottayam 686560, Kerala, India

**Keywords:** interpenetrating networks (IPN), morphology, macroporous polymers, electron spin resonance/electron paramagnetic resonance ESR/EPR-spin probe, delivery of bioactive molecules, novel food packaging

## Abstract

In this research, the porous polymer structures (IPN) were made from natural isoprene rubber (NR) and poly(methyl methacrylate) (PMMA). The effects of molecular weight and crosslink density of polyisoprene on the morphology and miscibility with PMMA were determined. Sequential semi-IPNs were prepared. Viscoelastic, thermal and mechanical properties of semi-IPN were studied. The results showed that the key factor influencing the miscibility in semi-IPN was the crosslinking density of the natural rubber. The degree of compatibility was increased by doubling the crosslinking level. The degree of miscibility at two different compositions was compared by simulations of the electron spin resonance spectra. Compatibility of semi-IPNs was found to be more efficient when the PMMA content was less than 40 wt.%. A nanometer-sized morphology was obtained for a NR/PMMA ratio of 50/50. Highly crosslinked elastic semi-IPN followed the storage modulus of PMMA after the glass transition as a result of certain degree of phase mixing and interlocked structure. It was shown that the morphology of the porous polymer network could be easily controlled by the proper choice of concentration and composition of crosslinking agent. A dual phase morphology resulted from the higher concentration and the lower crosslinking level. This was used for developing porous structures from the elastic semi-IPN. The mechanical performance was correlated with morphology, and the thermal stability was comparable with respect to pure NR. Investigated materials might be interesting for use as potential carriers of bioactive molecules aimed for innovative applications such as in food packaging.

## 1. Introduction

Porous materials are the focus of much research due to a number of excellent properties and application areas [1,2]. Various applications need a unique set of pore characteristics of the biopolymer network since the different morphology of created pores contributes to numerous performances of created biopolymer networks [3]. Single network hydrogel matrices have numerous applications, but a limiting feature is their weak mechanical characteristics and slow response at swelling [4]. Thus, a great amount of research attention was recently drawn to the interpenetrating polymer networks (IPN). These are defined as network structures formed by polymerized and/or crosslinked polymers within the immediate vicinity of each other. The IPN strategy relies on tailoring hydrogel matrices with more widely controllable physical (water holding capacity), thermal and mechanical properties. A special polymer blend structure is formed by the mutual transformation and entanglement. This unique entanglement structure combines the favorable properties of each polymeric component and can improve the compatibility among polymer chains, increase network density, and bonding force [5]. Likewise, specific porous materials are created, finding application in many different fields. Semi-interpenetrating polymer networks (semi-IPNs) and full IPNs are considered as novel technologies for example, to immobilize biomolecules, for controlled drug release with reduced side effects when ingested, for separation and filtration, as catalysis supports, for template-assisted synthesis of nanomaterials, and for various food applications such as edible films, three-dimensional printing, loading/releasing properties of bioactive/aroma compounds, or fat replacers [6,7]. The strengthening mechanisms of IPNs were mainly attributed to the filling of polysaccharides in other biopolymers, the increased density of entangled network, and the possible interactions between individual networks.

IPN hydrogels can be created from various polysaccharides such as chitosan and its derivatives for thermo- and pH-responsive matrices. In another example, cross-linking gelatin with rice leftovers, resulted in an IPN with better water resistance, oil resistance and improved thermal properties [4]. Moreover, acrylic acid grafted starch in the presence of poly(methacryloyl oxyethyl ammonium chloride) results in improved swelling capacity, cellulose IPN is used for packaging applications, silk sericin with poly(N-isopropylacrylamide and poly(methyl methacrylate) for a rapid pH-responsive hydrogel, etc. [8]. Poly(methyl methacrylate) (PMMA) is a non-toxic, lightweight and biocompatible thermoplastic polyester, mostly used for drug delivery and dental applications that makes it also a good candidate to be used as food contact material [9]. Recently, it was used in multicomponent antimicrobial active food packaging [10]. Polyisoprene (PI) is a rubbery polymer, either naturally derived from the sap of rubber trees (e.g., *Hevea brasiliensis*) or produced by polymerizing petroleum-derived raw material in a synthetic version. It can be used for food packaging applications in the form of films and tapes [11]. In controlled releasing systems, it was used for production of buccal patches for oral delivery of drugs [12].

To date, IPNs were produced either by sequential polymerization, simultaneous polymerization, or by latex blending [13,14]. During the synthesis, most IPNs form immiscible compositions. This results in separation at some synthesis steps and does not allow molecular interpenetration of the networks. Due to their interlocking phase configuration, there is a limited phase separation, with the domain dimensions ranging from hundred to tens of nanometers [15,16]. An IPN synthesis involves intimate mixing of crosslinked polymers, which makes it particularly interesting. A proper selection of components, their ratios and crosslinking level of the first formed phase significantly contributes to the morphology of IPNs, that is important for generating tailored types of porous materials. Semi-IPNs often create the dual phase continuity that result in crosslinked porous materials with excellent properties [17]. Although semi degradable block copolymers are ideal precursors for the preparation of ordered mesoporous thin polymer films, they are not suitable for macroscopic samples. Literature data demonstrate the effectiveness and versatility of removing the un-crosslinked PMMA from a natural rubber/PMMA based elastic semi-IPNs. This is presented as an alternative and straightforward strategy for creating macro to mesoporous networks from elastic matrix. This approach involving the semi-IPN system may provide an easy method for controlling the morphology associated with porous networks. To our knowledge, no studies have been published on porous materials prepared from semi-IPNs with an elastic component as its matrix. Even though different morphologies of block copolymers have been well studied, the phase shape and phase continuity in IPN’s are harder to predict [18]. There are two thermodynamic theories for sequential IPNs, both predicting spherical domains of polymer II (second phase) and, hence, discontinuous polymer II phase [19]. However, studies revealed that dual phase morphology is attainable in the case of sequential IPNs [18,20,21,22]. A better understanding of polymer mixing on molecular level and phase morphology of produced blend can be done by studying the viscoelastic properties and polymer chain dynamics using electron spin resonance (ESR). In the literature, it was reported that ESR spin probes follow different environments in a given sample and are, therefore, associated to the dynamics of the host polymer [23,24,25,26,27,28,29]. 

For example, mesoporous networks can be designed from such IPNs, where one subnetwork is degradable under specific conditions and the other is stable under same conditions. Commonly used selective degradation methods are hydrolysis [30], electron beam radiation [31], UV radiation [32] and simple dissolution using selective solvent [33]. In recent years, much progress has been made towards creation of nanoporous polymers with defined porosity [34,35,36,37]. 

In this study, firstly, the semi-IPNs were prepared from PMMA and polyisoprene. Secondly, porous structures were also prepared. A detailed characterization of structure, mechanical and thermal properties was done. The morphology (molecular weight, composition and crosslink density) of semi-IPNs and porous structures was determined by scanning electron microscopy (SEM), and microbeam small angle X-ray scattering (SAXS) analysis. The viscoelastic behavior and miscibility were studied by Dynamic Mechanical Analysis (DMA) and spin probe ESR.

## 2. Materials and Methods

### 2.1. Materials and Reagents

High molecular weight (M_w_~500,000–900,000) polyisoprene (natural rubber-NR) was supplied by the Rubber Research Institute of India (RRII) and low molecular weight polyisoprene (*M*_w_~39,000) was purchased from Aldrich (St. Louis, MO, USA) (Figure 1). Dicumyl peroxide (DCP), 99% and sulphur monochloride were used as crosslinking agents and were purchased from Aldrich. Prior to use, methyl methacrylate (MMA) (Aldrich) was distilled under vacuum. Azobisisobutyronitrile (AIBN) (Aldrich) was purified by recrystallization from methanol.

### 2.2. Sample Preparation 

#### 2.2.1. Semi-Interpenetrating Polymer Networks (Semi-IPNs)

Crosslinked sheets of polyisoprene with the thickness of 1–2 mm were weighted and kept immersed in a homogeneous mixture of methyl methacrylate and AIBN (0.7 g per 100 g of MMA). In order to make different formulations (weight percentages of PMMA), the NR sheets were swollen at different time intervals. The method was adapted from previous work [38] and further steps were followed: (1) the swollen samples were stored at 0 °C for several hours in order to obtain the equilibrium distribution of MMA monomer in the matrix; (2) the swollen networks were then heated for 6 h at 80 °C and for 2 h at 100 °C in a MMA atmosphere to complete the polymerization of MMA (*M*_w_(PMMA) = 733,000) and PDI = 3.09 as measured by gel permeation chromatography (GPC). In a final step, (3) the samples were placed in a vacuum air oven to remove the unreacted MMA from the semi-IPN. The final semi-IPN sample was weighed, and its composition was determined. The blends of NR and PMMA were prepared by mixing both components in toluene.

#### 2.2.2. Porous Semi-IPN

Prepared semi-IPNs were kept immersed in a mixture of cyclohexane and sulphur monochloride (0.8, 2 or 4 mL per 100 mL of cyclohexane) for 24 h to heavily crosslink NR matrix, or in other words until the matrix become rigid. The excess sulphur monochloride was removed by carbon disulphide.

The semi-IPNs were extracted with chloroform for 60 h at 70 °C in a Soxhlet apparatus. Extracted networks were then dried under vacuum. The extraction of linear PMMA from semi-IPNs thus led to the formation of residual porous NR networks.

The general symbol for semi-IPN samples was ^a^NRM_b_, where “^a^” denotes the percentage of crosslinker in NR (0.8, 2 or 4%), and “_b_” the percentage of PMMA. ^a^NRM_b_ symbol was used for the samples containing a low molecular weight NR, and an absence of “a” in the symbol denoted a blend. Semi-IPN samples were made with the following ratios of NR and PMMA: 80:20 (NRM_20_), 75:25 (NRM_25_), 65:35 (NRM_35_), 50:50 (NRM_50_), 45:55 (NRM_55_), 40:60 (NRM_60_) and 35:65 (NRM_65_). For the creation of porous structures, the composition should be close to 50:50. Lower molecular weight PI tends to enable the generation of continuous pores rather than isolated ones, and those samples were marked with letter “L” (for example NRLM).

### 2.3. Structural Analysis

#### 2.3.1. Scanning Electron Microscopy (SEM)

The scanning electron microscopy (SEM) analysis was used to study the structure morphology of prepared samples. Analyses were performed on JEOL JSM 6320F field emission gun scanning electron microscope. The samples were cryofractured and the surface was coated with platinum (Gatan HR 681 Ion beam coater).

#### 2.3.2. Small Angle X-ray Scattering Analysis (SAXS)

The SAXS analyses were used as a non-destructive method to study nanostructured forms of semi-IPNs [38]. The SAXS profiles were measured with the incident beam on the sample surface at 90°, and recorded on an area detector (in UMASS Amherst facility, Amherst, MA, USA). X-ray scattering was performed with Ni-filtered copper Kα-radiation from a Rigaku rotating anode, operated at 8 kW.

#### 2.3.3. Dynamic Mechanical Analysis (DMA)

The viscoelastic properties were studied using the DMA analysis (Rheometrics MK II DMA, Piscataway, NJ, USA) [23]. Measurements were taken in a tensile mode with frequency of 1 Hz and at a heating rate of 3 °C/min in the temperature interval from −85 to 170 °C.

#### 2.3.4. Spin Probe ESR Spectroscopy

The ESR spectroscopy was used for a better understanding of phase morphologies in produced blends. Analysis was conducted on a Varian E-109 spectrometer (operating at 9.3 GHz) paired with a Bruker ER 041 XG microwave bridge and a Bruker ER 4111 VT temperature unit. The nitroxide free radical, 4-hydroxy-2,2,6,6-tetramethylpiperidine-1-oxyl (TEMPOL), was used as a spin probe. Procedure given by [23] was followed with some modifications. In order to incorporate the probe molecules (at 0.15 wt.%), the semi-IPN samples were swollen in toluene probe solution. The measured data were processed using the EW (EPRWare) software. Based on the signal-to-noise ratio, the number of accumulations ranged from 2 to 5.

The ESR spectra were simulated using a NLSL spectral fitting program based on the Stochastic Liouville Equation (SLE). This was done in order to determine the relative proportions of slow and fast components in ESR spectra. The NLSL program uses a modified Levenberg–Marquardt minimization algorithm to calculate the optimal fit to experimental spectra. For each component, the initial spectral fits were obtained by changing parameters for rotational diffusion rate and isotropic Gaussian line broadening. By adjusting the orientation potential coefficients, the spectral fits were further refined.

### 2.4. Mechanical and Thermal Analysis

Mechanical properties were measured on Shimadzu AG II universal testing machine according to ASTM D-80 standard and adapted from [23].

Thermal properties were measured on TGA 2950 (TA Instruments Inc., New Castle, DE, USA) operating at: heating rate of 10 °C/min, temperature interval: 30–600 °C in N_2_ atmosphere. The glass transition temperatures (*T*_g_) of semi-IPNs were determined using DSC Q100 (TA Instruments Inc.) operating in N_2_ atmosphere, in temperature interval from −125 to 200 °C and at a heating rate of 20 °C/min.

## 3. Results and Discussion

### 3.1. Structure and Morphology of Semi-IPN

#### 3.1.1. Effect of Composition and Crosslink Density of Natural Rubber (NR) on the Structure and Morphology

The system’s morphology is often responsible for material properties, so solid state organization becomes crucial when designing multicomponent polymers with tailored properties. In this work, the semi-IPNs with PMMA fractions varying from 20 to 65 wt.% were analysed. Results on structure and morphology, before and after the removal of PMMA, are given in Figure 2 and Figure 3, respectively. The SEM pictures of blends are given in Appendix A. In the semi-IPN prepared with 20 wt.% of poly(methyl methacrylate), the poly(methyl methacrylate) phase was greatly dispersed in the continuous matrix made of natural rubber. This revealed the typical sea-island morphology (Figure 2a). According to the literature, the mechanical relaxation spectroscopy of this type of sample showed only one glass transition corresponding to NR phase [38]. The sample containing 35 wt.% of PMMA (Figure 2b) was also in a dispersed phase with a slightly larger domain dimension compared to that of 20 wt.%. Further increase of PMMA fraction strongly influenced the matrix structure seen as a shift from the sea-island to the dual phase morphology (Figure 2c). This indicated that polyisoprene, which was synthesized first, formed the matrix with NR as a major continuous phase and tended to control the morphology.

After the PMMA removal, porous structures remained in all samples. The porous structures in sample with 55 wt.% of PMMA (Figure 3a) ranged from 70 to 100 nm. This was an indication that before the removal, the PMMA phase has formed clusters of large spherical domains that were enveloped by small domains. The existence of the smaller pores was attributed to the crosslinking of the NR phase. The pore sizes were found to increase with increasing the amount of PMMA (Figure 3c). The nanometer-sized morphologies of IPNs were resulting from the crosslinking effect, that was limiting the polymer toughening during the polymerization and phase separation [39]. For all samples, the structuring of a nanometer sized domains was followed by an enhanced interphase mixing. Previously reported results of DMA analysis of the semi-IPN showed an inward shift of *T*_g_’s of both components, which might have be a sign of enhanced mixing [38]. This was also a confirmation of the finer morphology observed in SEM images in the present study. The increase in crosslink density of NR led to an overall decrease in the size of PMMA domains (as could be seen in Appendix A vs. Figure 3a). More interestingly, higher crosslinking resulted in a finer structure with greater regularity, as given in Figure 3a,b. SEM images showed the presence of vast number of PMMA domains with sizes approximately ranging from 30 to 70 nm. 

The small angle X-ray scattering (SAXS) profile of the semi-IPN ^4^NRM_50_ sample is given in Figure 4. From the position of the first-order scattering maxima, it was possible to calculate the PMMA inter-domain distance for this highly crosslinked semi-IPN. In theory, the inter-domain distance (D) is proportional to the Bragg spacing a = 2π/q_max_ and therefore inversely proportional to the scattering vector maxima (q_max_). Then, the relation between D and a was considered as a domain structure dependent [40]. The inter-domain distance for PMMA, derived from the SAXS data, was found to be ~89 nm. Here again, a higher crosslinking in NR resulted in the development of a nanometer-sized morphology. The types and the extent of crosslinking influenced the exact appearance in the fine structure that is usually a characteristic of the morphology of semi compatible materials.

The nodular structures were formed when the content of crosslinkers was decreased (as for ^0.8^NRM_65_ sample, Figure 3c). The PMMA phase has formed interconnected spherical nodules in the continuous NR phase. The characteristic morphology observed in this case showed a transition from the nodular structure to dual phase morphology, although the boundaries between these two structures were obscure and not regular. In the case of simple NR/PMMA blend, without the addition of crosslinkers (Appendix A), the transition from nodular structure to dual phase morphology was observed even at lower concentration of PMMA. This finding clearly showed the profound influence of crosslinking protocol in the ultimate development of different blend morphologies. Further increase of PMMA amount in the blend (Appendix A) led to an increase in dimensions of both NR and PMMA domains to the range of 1 μm. However, an existence of numerous small island-type NR domains with dimensions around 100 nm or lower was also observed. It is possible to suppose that these small islands might correspond to NR domains formed of the sol component (low molecular weight polyisoprene chains) that could have partially remained after the PMMA extraction in chloroform.

#### 3.1.2. Effect of Molecular Weight on the Structure and Morphology

During the semi-IPN synthesis, the phase that was firstly formed had a control on the morphology. Therefore, the molecular weight of the firstly formed NR phase also had a great impact on determining the size and shape of PMMA phase. It was previously found that the molecular weight influenced the matrix viscosity, so it should have a direct influence on the size of the dispersed particles [41,42,43,44]. It was also found that the addition of low molecular weight polyisoprene matrix (NR) at 50 wt.% led to the dual phase morphology (Figure 2c,d). So, in low molecular weight semi-IPN, the PMMA phase was highly occluded, and domains were large. As the *M*_w_ of NR matrix increased, both the PMMA particle size and the amount of occluded material decreased. The high molecular weight natural rubber maintained to form matrix even at 65 wt.% of PMMA (Figure 3c). In this sample, the higher stress levels generated with high *M*_w_ of NR led to a more continuous matrix and produced smaller particle sizes of the PMMA phase in comparison with the low molecular weight semi-IPN.

#### 3.1.3. Porous Structures from Semi-IPNs and Blends

The removal of PMMA phase resulted in a highly continuous porous material. Pore sizes were ranging from 100 nm to 250 nm for semi-IPN (for 50 wt.% and 35 wt.% samples, Figure 2d and Figure 2b, respectively). This showed the effective role of PMMA in creation of a spongy template. Since the *T*_g_ of the matrix was below the room temperature (as explained later in Section 3.5), the matrix was heavily crosslinked with sulphur monochloride before the removal of PMMA in order to prevent the collapse of pores. The pore size was strongly dependent on the crosslinking level of NR and concentration of PMMA. The pore size obtained by this protocol was greater than those that were possibly formed from block copolymers. A higher amount of crosslinker seemed to reduce the pore sizes, but this was not completely favorable because the connectivity between the pores could not be maintained in all cases. Indeed, for the successful utilization of porous material, the interconnected channels must be maintained. These demands control of the crosslinking level of NR matrix and concentration of PMMA. In developed materials, by increasing the wt.% of PMMA, pore sizes were higher too. So, in the case of NR/PMMA blend with 65 wt.% of PMMA, pore sizes were in the range of 1 μm (Appendix A), while for 50 wt.%, this range was from 150 to 200 nm (Appendix A). The addition of DCP crosslinkers already at 0.8 wt.%, decreased the pore size of blend with 65 wt.% PMMA to the range from 100 to 200 nm (Figure 3c). Thus, porous structures developed from semi-IPNs offer a simple morphology control of the porous network by carefully adjusting the above-mentioned parameters (crosslinking level and PMMA concentration). 

### 3.2. Results on Storage Modulus (Dynamic Mechanical Analysis) 

The DMA scans of the semi-IPNs are shown in Figure 5. In the NR sample, the storage modulus was sharply decreased around −70 °C. It was attributed to its α-relaxation process. The PMMA also showed a sudden drop in modulus in its glass transition region. When the storage modulus of the semi-IPNs were considered, it was seen that below the *T*_g_ of NR (close to −70 °C), all systems had better modulus values, since both NR and PMMA phases were in the glassy state. At around −50 °C, the NR became elastic and modulus behavior was predominantly due to the glassy PMMA phase. At around 110 °C, the PMMA also lost glassy nature and the modulus of samples dropped. Then, semi-IPNs showed two relatively sharp transitions, characteristics of incompatible materials. For semi-IPNs with low PMMA concentration, high temperature transition corresponding to PMMA could not be clearly seen during the DMA scans. This was because the semi-IPNs with less than 40 wt.% of PMMA developed a sea-island morphology, in which the NR was forming the matrix and the PMMA was highly scattered throughout the matrix. Theoretically, the modulus strongly depends upon the continuity of the phases. So, the transition corresponding to more continuous phase shows dramatic decrease in modulus. The presence of PMMA was dominant inside the NR network at low concentrations and in the dispersed state it did not influence the storage modulus of semi-IPN. As the concentration of PMMA was increased, a shift from the sea-island to the nodular structure morphology was observed, as discussed before (Figure 2). This nodular structure region obtained with the composition around 55 wt.% showed a better mechanical characteristic among semi-IPNs studied. Considerable increase in storage modulus could be seen in the sample with 50 wt.% compositions. From magnitudes of the two transitions, it can be seen that both phases were continuous. Decrease in slopes of both transitions along with the increase in modulus, showed the effect of close interpenetration of linear PMMA chains into a more continuous NR network. As result, the matrix was more rigid. The degree of interpenetration was more pronounced in highly crosslinked semi-IPNs as shown in Figure 5. This interphase mixing was already reported in the literature [38]. The highly crosslinked semi-IPN, the sample ^4^NRM_50_, took an intermediate position when compared to pure homopolymers and showed higher storage modulus values at all temperatures when compared to other semi-IPNs. The increase in storage modulus was due to some degree of phase mixing obtained by higher crosslinking of the first formed NR phase.

During the polymerization of PMMA, when the crosslink density of NR network was lower (*M*_w_ of crosslinks was high), the growing PMMA chains that were expanding, pushed apart the loosely bounded NR chains. As a result, semi-IPN with highly separated phases was formed. However, when the crosslink density of NR matrix was higher, the position of NR chains was hardly changed, enabling the PMMA chains to grow and interpenetrate into the already existing NR network. As a consequence, some phase mixing occurred.

Another interesting observation in the case of heavily crosslinked sample (^4^NRM_50_) was that its storage modulus was preserved even after increasing temperatures to those of the PMMA softening. The other three semi-IPNs showed a drop in modulus at 150 °C to the value similar to that of pure NR (Figure 5). This was unusual and interesting behavior of the highly crosslinked semi-IPN. In this case, the phase that was formed first was in the form of a 3D network with a higher crosslink density. During the polymerization of MMA monomer within the above three-dimensional network, the PMMA chains had to grow interpenetrating through the dense NR network chains in random directions. This resulted in the development of two conditions in the system. The interpenetration of PMMA chains through dense NR networks led to mixing between the two components to a certain degree, which resulted in an enforced miscibility. Since the PMMA chains grew in random directions, inside the dense NR network, this random orientation might have trapped most of the stiff PMMA chains inside the three-dimensional NR network. The development of the aforementioned two conditions along with the 50 wt.% PMMA content tended to make this semi-IPN to follow the modulus of PMMA at higher temperatures as seen in Figure 5. Here, the NR network chains were forced to follow the motions of the stiff PMMA chains and so the entire matrix showed a considerable degree of PMMA behavior.

Nanometer-sized morphology was not observed in the ^2^NRM_35_ sample that showed a certain degree of phase mixing. Here, the lower crosslinking in NR (2 wt.% of crosslinker) along with the absence of crosslinks in PMMA resulted in lower degree of interlocking. Thus, cooperative motions did not take place after the softening point of PMMA. However, the same compositions of full IPNs retained the storage modulus value up to 200 °C [23]. This clearly shows the efficiency of crosslinking of the second component in the interpenetration between both phases.

The drops in modulus that occur in each transition depend not only upon the composition, but also upon the relative continuity of two phases. In all semi-IPNs, a dramatic decrease in modulus was seen after the NR transition, which clearly indicates that NR formed the most continuous phase in all cases. Generally speaking, the polymer that is synthesized first, tends to be more continuous and in most cases forms matrix. In all samples in this study, the elastomeric NR network was synthesized first, then plastic component monomers were allowed to be swollen in, and in the last step they were polymerized. Therefore, the chains of the first NR network would always be strained to a certain extent and the linear chains of the second phase would be more nearly at equilibrium.

### 3.3. ESR Spectroscopy and Spectral Simulations

The ESR spectra of 65/35 NR/PMMA semi-IPN (^2^NRM_35_) and highly crosslinked 50/50 semi-IPN (^4^NRM_50_) samples at 75 °C showed a very interesting behavior that can be directly correlated to the effect of miscibility. Similar behaviors were previously seen in semi [38] and full IPNs [23] at low concentrations of PMMA (less than 35 wt.%). Figure 6 shows the ESR spectra of the two semi-IPNs at 75 °C. In contrast to less crosslinked sample, the 35 wt.% semi-IPN, the spectrum of highly crosslinked semi-IPN with 50/50 composition had a larger proportion of a mobile component. In both of these semi-IPNs, the NR phase was more continuous and the PMMA phase was dispersed throughout the NR matrix. Therefore, the lower amount of mobile fraction of the 35 wt.% sample resulted from their miscibility.

The estimation of quantity of the fast as well as of the slow motion components in the semi-IPNs are given in Table 1. The semi-IPN with lower PMMA concentration had a lower amount of fast component when compared to the samples with higher PMMA content. This was in accordance with the DMTA and SEM analysis, from which it was shown that the IPN with low PMMA concentration (less than 35 wt.%) had the maximal miscibility of all the studied samples (semi and full IPNs) [23,38]. The ESR observations confirmed the higher miscibility at PMMA concentrations lower than 35 wt.%. The measured data showed that the miscibility had more important influence than the crosslinking effect. Then, as a consequence, the NR chains that were more mobile had become stiffer with limited motion at PMMA concentrations lower than 35 wt.%. Thus, the effect of miscibility was responsible for the restrictions of the molecular motion of the highly mobile NR chains. This was reflected as the lower amount of fast (mobile) component in the semi-IPN having 35 wt.% PMMA at 75 °C in comparison with the 50 wt.% semi-IPN and measured at the same temperature. The α-relaxation of NR in the 35 wt.% semi-IPN sample showed a shift to higher temperature. Moreover, the relaxation was widening, and a shoulder appeared on the high temperature side of NR transition [38]. The broadened NR relaxation in 35 wt.% semi-IPN was much more noticeable to the higher temperature side when compared to the highly crosslinked (^4^NRM_50_) samples. This means that in sample with 35 wt.% of PMMA, the matrix was strengthened to higher extent than the other as noticed in the ESR observation.

### 3.4. Mechanical Properties

The mechanical properties of the semi-IPNs were plotted as a function of the PMMA wt.% and shown in Figure 7a–c. A progressive change from rubbery to plastic behavior was observed along to the increased concentration of PMMA. A significant change was observed in samples with 50 wt.% PMMA. The tensile strength and the modulus were greatly correlated to morphology, showing a noticeable change near domains where a nodular structure morphology was formed. Particularly, the tensile strength and the modulus were lower in the sea-island region, with a sharp increase when morphology changed to a nodular structure with a sudden decrease as the dual phase morphology was appearing. The tensile strength also increased significantly when the PMMA concentration was higher than 30 wt.%, because the two constituents started to form co-continuous phases (i.e., nodular structures). Elongation at break was decreasing gradually with increasing the concentration of PMMA due to a low elongation and a brittle nature of the PMMA phase. The results obtained suggested that the region containing a nodular structure (with higher tensile strength and modulus) also enhanced the tensile properties of the semi-IPNs. Highly crosslinked semi-IPN showed an increase in modulus, but had low tensile strength and elongation at break.

### 3.5. Thermal Stability

The results of thermal analysis are given in Figure 8 and Figure 9. From the TGA profiles (Figure 8), it can be seen that the presence of linear PMMA in semi-IPNs had no significant influence on the thermal stability of NR/PMMA up to 400 °C. At this end, about <40% of sample weights still remained. This was comparable to the pure crosslinked NR (^2^NR) sample. In the final degradation stage, for temperatures above 400 °C, a slight shift could be seen for semi-IPNs with temperatures needed for full degradation somehow lower than that for pure NR. 

From the DSC profiles (Figure 9), it was evident that the NR showed only one transition around −58 °C, while two glass transitions (*T*_g_) for all semi-IPNs were noticed (at −58 and 117 °C). The second *T*_g_ was attributed to the PMMA. Peak broadening was a sign of a slight variation in the local glass transition temperatures, that was due to the increased mobility of PMMA chains as a consequence of mixing flexible NR chains near stiff PMMA. The second *T*_g_ was barely visible from the elongation, and this elongation was a consequence of the increased mobility of PMMA due to the proximity of the flexible NR chains. Indeed, the two glass transitions are characteristic for phase separated multicomponent matrices (as was the case for tested semi-IPNs). The highly crosslinked semi-IPN and low molecular weight semi-IPN had a slight inward shift in NR’s *T*_g_ which was an indication of mixing as observed in the SEM analysis (Figure 3).

## 4. Conclusions

A detailed characterization of NR/PMMA semi-IPNs with SEM, SAXS, DMA, spin probe ESR, mechanical analysis, TGA and DSC is presented. The effects of crosslinking, composition and molecular weight were carefully analyzed in order to study the compatibility in semi-IPNs. Factors that determine the morphology of porous structures from semi-IPNs were also discussed. Lower concentrations of PMMA (below 35 wt.%) showed a sea-island morphology. With increasing the concentration of PMMA to 50 wt.%, the system shifted to nodular structure morphology. This region showed maximum mechanical performance in semi-IPNs. Higher crosslinking in the NR phase led to the development of nanometer-sized morphology as a result of better interpenetration and mixing between two phases. The crosslinking density of the first formed phase was determined as the key factor responsible for controlling the miscibility and morphology of NR/PMMA semi-IPNs. It was found that the interlocked structure developed due to the heavy crosslinking, makes semi-IPN to follow the modulus of PMMA at higher temperatures. Broad *T*_g_ of PMMA phase indicated fluctuations in local glass transition temperatures which arose due to certain degree of phase mixing. Compatibility of semi-IPNs was found to be more efficient when the PMMA content was less than 40 wt.%. With 60 wt.% of PMMA, system showed a dual phase morphology. The system attained a dual phase morphology at a lower concentration of PMMA when the molecular weight of NR matrix was also reduced. This dual phase morphology (bicontinuous) was utilized to develop porous structures with good mechanical strength from elastic semi-IPNs by completely removing the linear PMMA by simple dissolution. Careful selection of crosslinking level, compositions and components was shown to provide a convenient way to control the morphology of the resulting porous structures. The system showed comparable thermal stability with respect to pure NR. One of the disadvantages of the porous structures developed in this work was its brittleness. The high brittleness can be reduced by lowering or fine-tuning the concentration of the sulphur monochloride. Further investigations with varying levels of crosslink density could provide insight into the adequate level of network density required to synthesize a stable and robust porous structure with mechanical integrity.

## Figures and Tables

**Figure 1 polymers-15-01353-f001:**
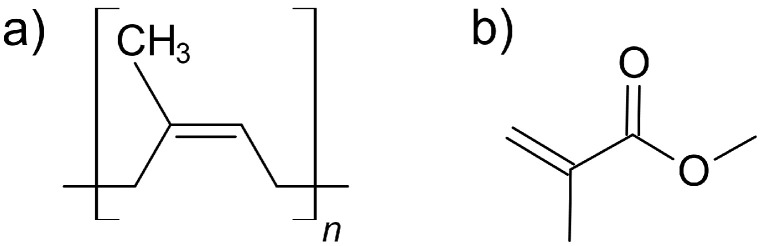
Chemical structures of (**a**) polyisoprene (NR) and (**b**) methyl methacrylate (MMA).

**Figure 2 polymers-15-01353-f002:**
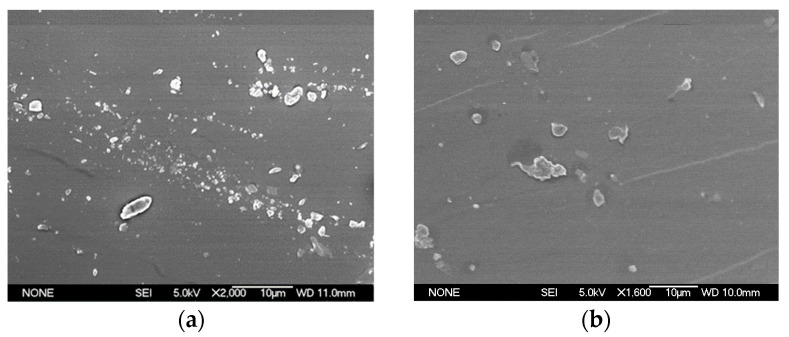
SEM images of various semi-IPNs: (**a**) ^2^NRM_20_; (**b**) ^2^NRM_35_; (**c**) ^2^NRM_55_; (**d**) ^4^NRM_50_. Photos taken in magnification from ×1600 to ×2000.

**Figure 3 polymers-15-01353-f003:**
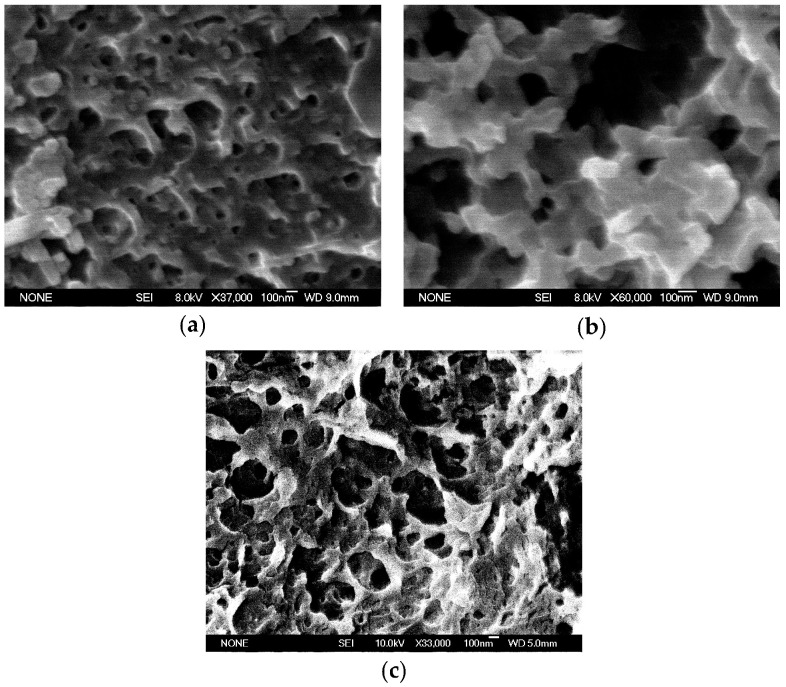
SEM images of porous semi-IPNs after the PMMA removal: (**a**) ^2^NRM_55_; (**b**) ^2^NRLM_50_; (**c**) ^0.8^NRM_65_.

**Figure 4 polymers-15-01353-f004:**
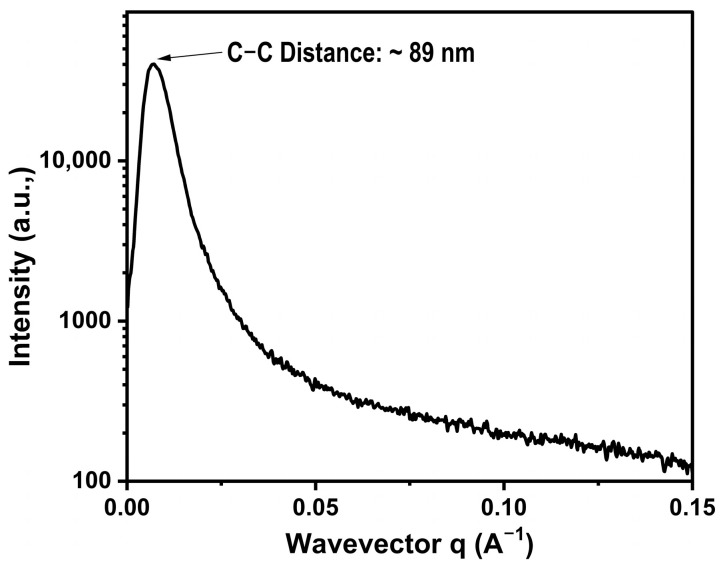
SAXS profile of the highly crosslinked ^4^NRM_50_ semi-IPN.

**Figure 5 polymers-15-01353-f005:**
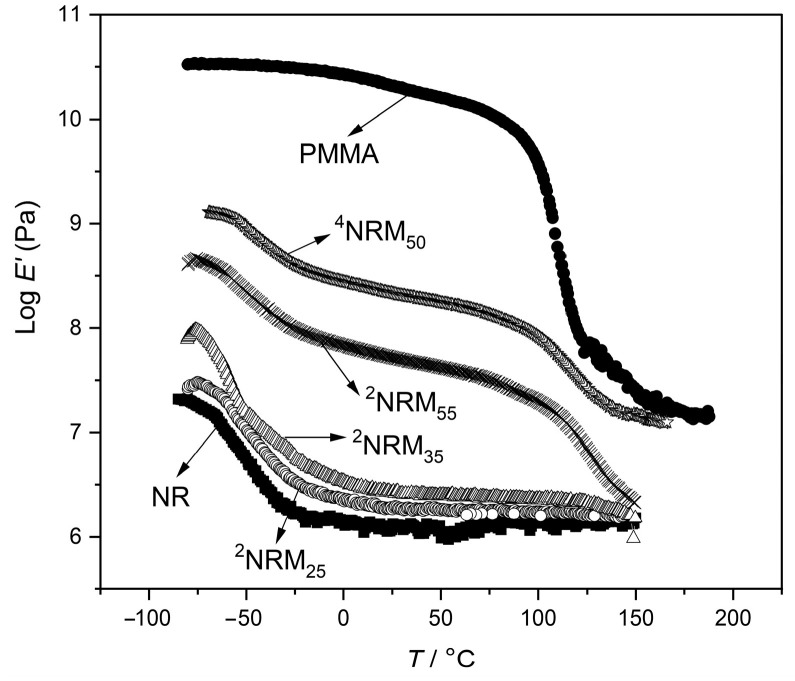
Storage modulus of semi-IPNs and homopolymers as a function of temperature.

**Figure 6 polymers-15-01353-f006:**
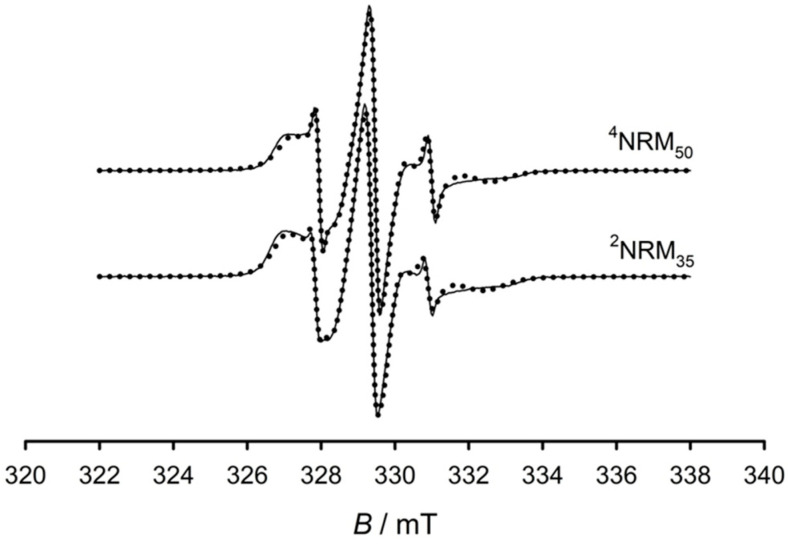
The ESR spectra of ^2^NRM_35_ and ^4^NRM_50_ semi IPNs at 75 °C.

**Figure 7 polymers-15-01353-f007:**
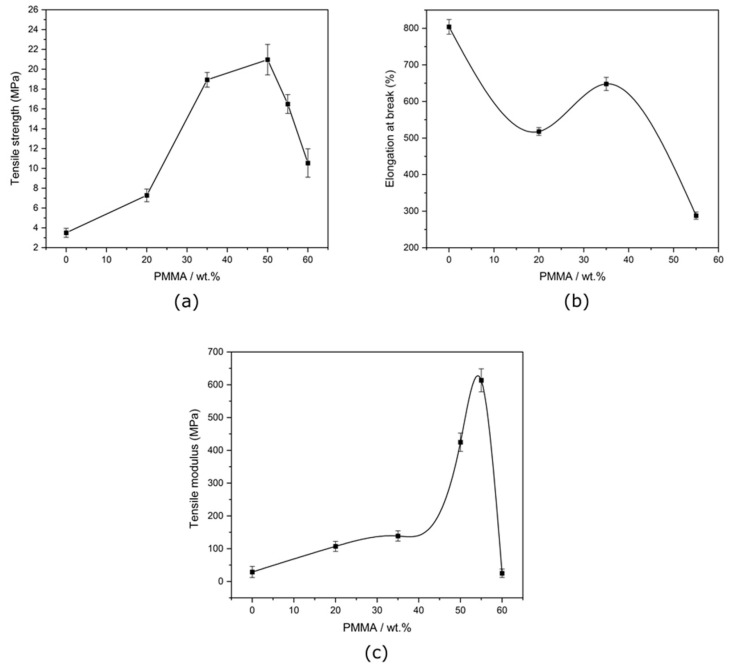
The effect of PMMA concentration in crosslinked semi-IPNs on (**a**) tensile strength, (**b**) elongation at break, and (**c**) tensile modulus.

**Figure 8 polymers-15-01353-f008:**
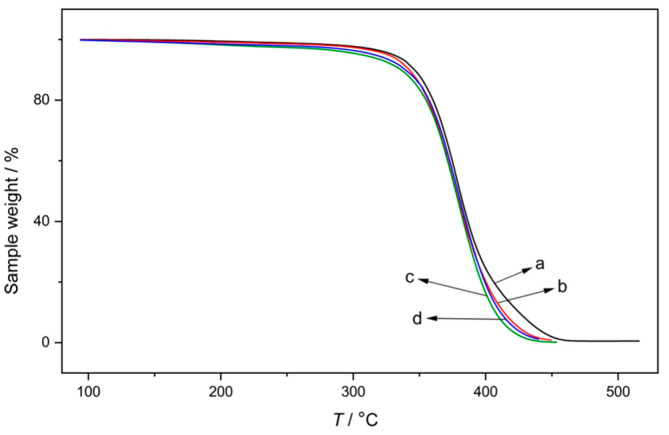
TGA curves of (**a**) crosslinked pure NR (^2^NR) and semi IPN samples, (**b**) ^2^NRM_35_, (**c**) ^2^NRM_55_, and (**d**) ^4^NRM_50_.

**Figure 9 polymers-15-01353-f009:**
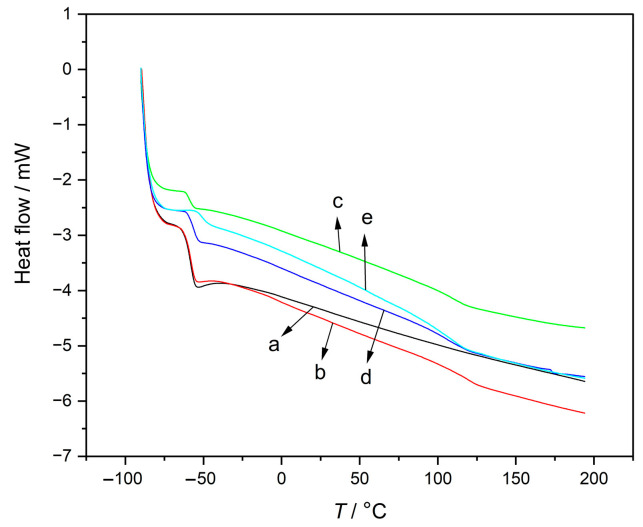
DSC curves of: (**a**) pure NR (^2^NR) and semi-IPNs (**b**) ^2^NRM_35_, (**c**) ^2^NRM_55_, (**d**) ^4^NRM_50_, and (**e**) ^2^NRLM_50_.

**Table 1 polymers-15-01353-t001:** Quantitative estimation of the amount of slow and fast components from the ESR spectra measured at 75 °C.

Sample	Slow Component/%	Fast Component/%
^2^NRM_35_	90.7	9.3
^4^NRM_50_	87.6	12.4

## Data Availability

Not applicable.

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
