# Peer review of "Phase Behavior of NR/PMMA Semi-IPNs and Development of Porous Structures"

_polymers, 2023, doi:10.3390/polym15061353_

Round 1
Reviewer 1 Report
The subject of presented manuscript is relevant and interesting. The wide range of used chemical and physical methods of complex polymer products seems to be impressive and adequate to the scientific goal. I would suggest to improve text by adding of chemical formula that could help to understand more clear the idea. The declared biomedical applications were not confirmed. Also list of references contains too many old reviews. English grammar should be carefully checked and corrected. My recommendation: accept after minor revision.
Author Response
Dear Editor, dear reviewers,
Thank you for your valuable comments.
Authors followed all the recommendations suggested by reviewers that are detailed below. In the Revised Manuscript file, all the changes are highlighted in yellow. Revised manuscript is thus significantly improved. Significant improvements in English language are done, and corrections are checked by an English writing and editing tool.
Some new references are also added so final number of references has been changed as well.
We hope that in this form article follows all your recommendations and that can be considered as acceptable.
Reviewer’s comments
Reviewer 1
The subject of presented manuscript is relevant and interesting. The wide range of used chemical and physical methods of complex polymer products seems to be impressive and adequate to the scientific goal. My recommendation: accept after minor revision.
Answer:
Authors thank to reviewers’ effort and provided comments. All suggestions are taken into account and the improved manuscript is provided.
Comment
I would suggest to improve text by adding of chemical formula that could help to understand more clear the idea.
Answer:
Chemical formula is added as new Figure (Figure 1).
Figure 1. Chemical structures of a) polyisoprene (NR) and b) methyl methacrylate (MMA).
Since the new Figure is added, the final number of Figures is 9.
Comment
The declared biomedical applications were not confirmed. Also list of references contains too many old reviews.
Answer:
The aim of this study wasn’t to confirm the application but this was concluded from the results as potential applications. Even though no many research was done within the field, the list of references is improved, and more recent references are added. However, not to mislead the reader the featured application was deleted from the abstract and introduction part.
- Udenni Gunathilake, T.M.S.; Ching, Y.C.; Ching, K.Y.; Chuah, C.H.; Abdullah, L.C. Biomedical and Microbiological Applications of Bio-Based Porous Materials: A Review. Polymers (Basel) 2017, 9.
- Du, M.; Lu, W.; Zhang, Y.; Mata, A.; Fang, Y. Natural Polymer-Sourced Interpenetrating Network Hydrogels: Fabrication, Properties, Mechanism and Food Applications. Trends Food Sci Technol 2021, 116.
- Raina, N.; Rani, R.; Khan, A.; Nagpal, K.; Gupta, M. Interpenetrating Polymer Network as a Pioneer Drug Delivery System: A Review. Polymer Bulletin 2020, 77, 5027–5050.
Comment
English grammar should be carefully checked and corrected.
Answer:
The manuscript was checked thoroughly and English language was improved.

Reviewer 2 Report
This work developed IPN from natural NR and PMMA, and determined the effects of molecular weight and crosslink density of polyisoprene on the morphology and miscibility with PMMA. The authors also prepared sequential semi-IPNs and investigated their viscoelastic, thermal and mechanical properties. The crosslinking density of the natural rubber was found to be the key factor that influenced the miscibility in semi-IPN. The degree of compatibility was increased by doubling the crosslinking level. Compatibility of semi-IPNs was more efficient with PMMA content less than 40 wt. %. Interestingly, the mechanical performance was correlated with morphology, and thermal stability was comparable with respect to pure NR. This material may be interesting for use as potential carriers of bioactive molecules aimed for innovative pharmaceutical applications and food packaging. The manuscript is in general well written and the method development process is described in detail. However, the following issues have to be addressed before this manuscript is suitable for publication.
1. The section Introduction should be reorganized, in order to provide the background information in a much more logical way, such as the research progress and application fields of porous materials, the production technology and degradation methods of interpenetrating polymer networks. Herein, 4-5 paragraphs are encouraged in this section.
2. In the sections 2.2, 2.3, and 2.4, the appropriate number of references should be provided, in order to help other research worker to follow your work.
3. Results have not discussed up to the mark in the sections 3.1.2, 3.1.3, 3.4, and 3.5. The results of this work should be compared with the reported research previously. Also, please provide the relevant literature.
4. Figure 2 and 6 should be re-combined, and some of your figures are suggested to move into the Supplementary Materials.
5. The limitations of the developed IPN should be discussed, and please look into your future work to improve its performance index.
Author Response
Dear Editor, dear reviewers,
Thank you for your valuable comments.
Authors followed all the recommendations suggested by reviewers that are detailed below. In the Revised Manuscript file, all the changes are highlighted in yellow. Revised manuscript is thus significantly improved. Significant improvements in English language are done, and corrections are checked by an English writing and editing tool.
Some new references are also added so final number of references has been changed as well.
We hope that in this form article follows all your recommendations and that can be considered as acceptable.
Reviewer 2
This work developed IPN from natural NR and PMMA, and determined the effects of molecular weight and crosslink density of polyisoprene on the morphology and miscibility with PMMA. The authors also prepared sequential semi-IPNs and investigated their viscoelastic, thermal and mechanical properties. The crosslinking density of the natural rubber was found to be the key factor that influenced the miscibility in semi-IPN. The degree of compatibility was increased by doubling the crosslinking level. Compatibility of semi-IPNs was more efficient with PMMA content less than 40 wt. %. Interestingly, the mechanical performance was correlated with morphology, and thermal stability was comparable with respect to pure NR. This material may be interesting for use as potential carriers of bioactive molecules aimed for innovative pharmaceutical applications and food packaging. The manuscript is in general well written and the method development process is described in detail. However, the following issues have to be addressed before this manuscript is suitable for publication.
Comment
- The section Introduction should be reorganized, in order to provide the background information in a much more logical way, such as the research progress and application fields of porous materials, the production technology and degradation methods of interpenetrating polymer networks. Herein, 4-5 paragraphs are encouraged in this section.
Answer:
The Introduction section is reorganized so the background information is provided in much more logical way, such as the research progress and application fields of porous materials in first two paragraphs, the production technology in the third and degradation methods of interpenetrating polymer networks in the 4th paragraph. New recent references are also added. Thus, the Introduction part is much more comprehensive.
New references:
- Udenni Gunathilake, T.M.S.; Ching, Y.C.; Ching, K.Y.; Chuah, C.H.; Abdullah, L.C. Biomedical and Microbiological Applications of Bio-Based Porous Materials: A Review. Polymers (Basel) 2017, 9.
- Du, M.; Lu, W.; Zhang, Y.; Mata, A.; Fang, Y. Natural Polymer-Sourced Interpenetrating Network Hydrogels: Fabrication, Properties, Mechanism and Food Applications. Trends Food Sci Technol 2021, 116.
- Raina, N.; Rani, R.; Khan, A.; Nagpal, K.; Gupta, M. Interpenetrating Polymer Network as a Pioneer Drug Delivery System: A Review. Polymer Bulletin 2020, 77, 5027–5050.
Comment
- In the sections 2.2, 2.3, and 2.4, the appropriate number of references should be provided, in order to help other research worker to follow your work.
Answer:
References are added to sections 2.2., 2.3., and 2.4.
Comment
- Results have not discussed up to the mark in the sections 3.1.2, 3.1.3, 3.4, and 3.5. The results of this work should be compared with the reported research previously. Also, please provide the relevant literature.
Answer:
Authors thanks for this remark. Indeed, the relation to presented results/figures was missing so it was added to the sections 3.1.2., 3.1.3., 3.4. and 3.5. Since no many similar research was done previously, at least not on the same polymer types, it was difficult to compare results with other studies, making importance and impact of the presented data.
Comment
- Figure 2 and 6 should be re-combined, and some of your figures are suggested to move into the Supplementary Materials.
Answer:
Authors thank for this comment. As suggested, some parts are moved to the Supplementary Materials. In Figure 2, photos d and e were moved to Supplementary, as Supplementary 1. As new Figure 1 is added to manuscript, then Fig. 2 became Fig. 3 in the new revised version. In our opinion recombining Figures 2 and 6 would completely change the structure of the whole manuscript so authors prefer to keep it like written. Combining structure and mechanical properties in the same figure would require the total reorganization of the written and discussed data.
Comment
The limitations of the developed IPN should be discussed, and please look into your future work to improve its performance index.
Answer:
Authors thank for the comment. The whole paragraph on disadvantages of NR/PMMA semi-IPNs is added to the conclusion section.
Disadvantages of NR/PMMA semi IPNs
One of the disadvantages of the porous structures developed in this work was its brittleness. The high brittleness can be reduced by lowering or fine-tuning the concentration of the Sulphur monochloride. Further investigations with varying levels of crosslink density could provide insight into the adequate level of network density required to synthesize a stable and robust porous structure with mechanical integrity.
